# One-Anastomosis Gastric Bypass and Hiatal Hernia: Nissen Fundoplication with the Excluded Stomach to Decrease the Risk of Postoperative Gastroesophageal Reflux

**DOI:** 10.3390/jcm11216441

**Published:** 2022-10-30

**Authors:** Antoine Soprani, Hortense Boullenois, Viola Zulian, Anamaria Nedelcu, Sergio Carandina

**Affiliations:** 1Department of Digestive and Bariatric Surgery, Geoffroy-Saint Hilaire Hospital, Générale de Santé (GDS), 75005 Paris, France; 2ELSAN, Saint Michel Hospital, Centre Chirurgical de l’Obésité (CCO), 83100 Toulon, France; 3Department of Digestive and Bariatric Surgery, Clinica Madonna della Salute, 45014 Porto Viro, Italy

**Keywords:** one-anastomosis gastric bypass, hiatal hernia, obesity, Nissen fundoplication

## Abstract

Introduction: The realization of an antireflux valve according to the Nissen technique during the operation of one-anastomosis gastric bypass (OAGB) may theoretically decrease the incidence of postoperative reflux in patients with hiatal hernia (HH). Material and Methods: In this retrospective study, we included all patients operated on between January 2015 and January 2019 for an OAGB associated with the creation of an antireflux Nissen valve (360-degree wrap). The patients included had type II or type III HH that had been diagnosed preoperatively or discovered intraoperatively. Results: Twenty-two patients were operated on during the period considered. The mean preoperative BMI was 40 ± 14 kg/m^2^. Five patients (22.7%) had a history of bariatric surgery. Typical symptoms of gastroesophageal reflux disease (GERD) were preoperatively present in four patients (18%), and HH was revealed preoperatively only in four patients; for all the other patients, the diagnosis of HH was made intraoperatively. The rate of early and/or late postoperative complications was 0%. The mean duration of follow-up was 23 ± 15 months. No dysphagia was reported during follow-up. Three patients presented with symptomatic GERD postoperatively, including one de novo. Mean BMI at the end of follow-up was 24 ± 3 kg/m^2^, and the % of total weight loss was 108 ± 30%. Conclusions: OAGB with a Nissen antireflux valve seems to be a safe and effective surgical technique and it could be an extra arrow in the surgeon’s quiver in the presence of HH in a patient scheduled for OAGB.

## 1. Introduction

Bile reflux is considered to be the Achilles’ heel of the one-anastomosis gastric bypass (OAGB) due to its frequency and the potentially carcinogenic effects of bile [1]. In the various series, the incidence of this complication varies from 0.6% to 10%, and various factors can favor its appearance [2]. Alongside technical errors, such as the creation of a too-short gastric pouch, there are preexisting conditions that determine the inability of the cardia to provide an adequate barrier to the reflux of liquid into the esophagus [3]. Previous restrictive bariatric surgery such as laparoscopic gastric banding (LAGB) or laparoscopic sleeve gastrectomy (LSG) has been shown to increase the incidence of biliary reflux (BR) in several studies [4,5,6,7,8]. Another factor frequently implicated in the genesis of gastroesophageal reflux disease (GERD) is the presence of a hiatal hernia (HH), regardless of its size [9]. In fact, by causing a laxity of the lower esophageal sphincter (LES), the hiatal hernia reduces the barrier functions of the esophagus-gastric junction against the reflux of the gastric contents. In current practice, the incidence of HH in the obese population is frequent, and the diagnosis is often confirmed intraoperatively [10,11]. The classical surgical treatment of HH is the association of the hiatoplasty with the creation of a full or partial antireflux valve using the fundus of the stomach [12]. Realization of an antireflux valve according to the Nissen technique during the operation of OAGB may theoretically decrease the incidence of postoperative reflux after OAGB in patients with obesity and HH. The purpose of this retrospective study was to evaluate on the one hand the feasibility and safety of the OAGB with a Nissen valve and, on the other hand, its effectiveness in preventing the postoperative appearance of BR.

## 2. Materials and Methods

In this retrospective study, we included all patients operated on between January 2015 and January 2019 for an OAGB associated with the creation of an antireflux valve according to the Nissen technique (360-degree wrap). All patients met the international criteria for the surgical treatment of obesity, and data pertaining to each patient, including demographic, preoperative, and postoperative clinical data, were collected.

At the time of the surgery, all the patients had been followed for more than 6 months in our obesity center by a multidisciplinary team involving a nutritionist, dietician, and psychologist, and all had achieved weight stabilization or weight loss during this follow-up. According to the recommendation of French authorities, all the patients underwent preoperative esophagogastroduodenoscopy with biopsies (EGD). In case of Helicobacter pylori infection, an appropriate antibiotic treatment was systematically administrated, and its eradication before surgery was verified with a breath test. In all the patients, EGD was completed with a preoperative upper gastrointestinal series (UGI).

The patients included in this retrospective series had type II HH (paraesophageal hiatal hernia) or type III HH (both paraesophageal hernia and sliding hernia, where the esophagogastric junction and the fundus are in the intrathoracic position) that had been diagnosed preoperatively or discovered intraoperatively. The absence of preoperatively symptomatic GERD as well as the removal of the adjustable gastric band during the same operation were not exclusion criteria. On the other hand, we excluded patients with a preoperative diagnosis of a large HH, independently of the type, especially if associated with the presence of severe GERD. Patients with a small (<2 cm) type I HH underwent OAGB with gastropexy, without Nissen fundoplication.

The surgical data collected were biliary loop length, associated procedure, operative time, hospital stay, intraoperative bleeding, early complication (at 30 days), rehospitalization, and late complication (during the follow-up period), while the data collected postoperatively were weight loss and the presence of symptoms of gastroesophageal reflux or dysphagia. Weight loss results were expressed as the change in BMI, percentage of total weight loss (%TWL), and percentage of excess weight loss (%EWL). The %TWL was calculated as [(preoperative weight − follow-up weight)/(preoperative weight) × 100]. The %EWL was calculated using the equation [(preoperative weight − follow-up weight)/(preoperative weight − ideal weight) × 100], where the ideal weight was considered to be equivalent to a BMI of 25 kg/m^2^. 

Continuous demographic variables and outcome variables were expressed as mean ± standard deviation. Categorical variables, in addition to complications, were reported as numbers and percentages.

### Surgical Technique

With the patient placed in the modified lithotomy position, OAGB was performed using a 5-port technique. After opening the lesser omentum and the phrenoesophageal membrane, the pillars of the diaphragm were completely dissected, and the hernial sac was resected after reduction of the hernial contents. The abdominal esophagus was dissected, taking care to respect the vagus nerves as well as the left gastric vessels (Figure 1). The hiatus was closed with 2–3 separate stitches of nonabsorbable sutures (Figure 2). A long and narrow gastric tube was created by applying one 60 mm horizontal linear staple at the angle of the lesser curvature, just proximal to the crow’s foot, and three to five vertical 60 mm staples toward the angle of His. A 36 Fr gastric bougie was used to size the creation of the new gastric tube. A bypass with an intestinal loop of 150 cm from the ligament of Treitz was made for patients with a BMI between 35 and 40, while for patients with BMI > 40 the loop length was 200 cm. The jejunal loop was pulled into position in an antecolic fashion, and an end-to-side gastrojejunal anastomosis (GJA) was performed with a linear 45 mm staple. The staple openings were closed with a resorbable running suture. A perioperative methylene blue dye test was performed to identify leakage. According to the Nissen technique, the 2.5–3 cm-long 360-degree wrap was created by passing the gastric fundus of the excluded stomach into the retroesophageal space (Figure 3). To reduce the risk of postoperative dysphagia, a wide mobilization of the fundus was essential: the posterior tuberosity artery and the last 2 short vessels should be systematically sectioned (Figure 1). A UGI series was performed on postoperative day (POD) 1, and, if negative for a leak, the patient started alimentation according to the recommendations given by the nutritionist. Discharge was authorized from POD 1 after clinical examination and verification of good food intake. All patients received a prescription for systematic postoperative multivitamin supplementation and 40 mg of a proton pump inhibitor (PPI) during the first 6 months after surgery. Anticoagulant prophylaxis was started on POD 1 and continued for the first postoperative month. Patient follow-up included an outpatient consultation at 1, 3, 6, 9, and 12 months, then twice a year. An UGI series was scheduled at 1, 3, and 5 postoperative years, while an EGD was scheduled at 3 and 5 postoperative years and in every case of severe postoperative GERD symptoms.

## 3. Results

Twenty-two patients were operated on for OAGB with a Nissen valve between January 2015 and January 2019. Table 1 summarizes the clinical characteristics of the patients preoperatively. The average preoperative age was 48 ± 10 years, and the mean preoperative BMI was 40 ± 14 kg/m^2^. Five patients (22.7%) had a history of bariatric surgery; in particular, four had previously undergone gastric banding (removed before conversion to OAGB), and one had had an intragastric balloon placed. Typical symptoms of GERD were preoperatively present in four patients (18%), and in three of these gastroscopy revealed signs of esophagitis stage B according to the Los Angeles classification. Hiatal hernia was revealed preoperatively only in four patients (18%); for all other patients, the diagnosis of hiatal hernia was made during the operation.

All operations were performed laparoscopically, and in no case was conversion to laparotomy necessary. The mean duration of surgery was 60 ± 20 min, and, in one patient, a cholecystectomy for preoperative symptomatic gallstones was associated. The biliary loop length was tailored to the preoperative BMI; in 11 patients, it was 150 cm, and in the other 11, 200 cm. The rate of early postoperative complications, as well as that of mortality, was 0%. No patient was rehospitalized during the 30 days following the surgery (Table 2).

The mean duration of follow-up was 23 ± 15 months. No dysphagia was reported during follow-up. No signs of HH recurrence were evidenced in the UGI control series performed at 1 and 3 years after surgery. Three patients presented with symptomatic GERD postoperatively, including one de novo. Two of them had a history of AGA complicated by symptomatic GERD with esophagitis. Reflux symptoms persisted despite removal of the band with the presence of erosive esophagitis before the Nissen-BPGO. The diagnosis of type II HH had been made preoperatively by UGI series and gastroscopy. Since the OAGB with a Nissen valve, the symptomatology of GERD has improved and remains well controlled by long-term PPIs. For one patient, gastroscopy and UGI were normal, and she presented no symptoms of GERD preoperatively. Intraoperatively, there was a type III HH. She developed a daily BR at 9 months from OAGB (rate of late postoperative complications: 4.5%). Mean BMI at the end of follow-up was 24 ± 3 kg/m^2^, the %TWL was 49.1 ± 9.3, and %EWL was 108% ± 30.

## 4. Discussion

In a recent study, Werapitiya et al. demonstrated the efficacy of creating a Nissen valve to treat refractory biliary reflux after OAGB, avoiding the classic conversion to gastric Roux-en-Y bypass [13]. In that study, 10 out of 12 patients had an HH. Conversely, in the present study, all the patients still had to undergo OAGB, and the discovery of the HH was made especially during the operation. The presence of an HH is known to be an important factor in the pathogenesis of GERD, inducing incontinence of the cardia. In the presence of an HH, fundoplication remains the treatment of choice [14,15]. In our series, the use of the fundus pouch of the excluded stomach to perform a modified Nissen is technically feasible, with a slightly longer operating time compared to standard OAGB (60 min vs. 45 min). Moreover, there was no conversion to laparotomy or acute postoperative complication, and all the patients were discharged the day after the operation. It is interesting to note that in this series no case of valve necrosis was reported despite the systematic section of the gastric posterior tuberosity artery and the last two short vessels of the greater gastric curvature. Moreover, no cases of postoperative dysphagia were recorded. In our opinion, this is probably due to the extensive mobilization of the gastric fundus, which allows the creation of a 360-degree floppy valve around the gastric pouch, without the consequent risk of traction on the gastric tube.

During the 2019 modified Delphi consensus on OAGB, the 52 attendee experts were unable to decide on the indication of OAGB in the event of a patient with morbid obesity associated with HH, regardless of its size [16]. First of all, the difficulty in diagnosing this type of hernia must be emphasized. Patients most often present with atypical symptoms (thoracic pain, dyspnea, regurgitation, dyspepsia). For type II and III HH, the classic reflux symptomatology is much less frequent than in the case of type I HH (sliding HH), where GERD is associated in 10–15% of cases. Imam et al. evaluated the interest of gastroscopy and upper gastrointestinal series performed during the preoperative workup [17]. The authors concluded that these two examinations had poor sensitivity in the detection of HH except for symptomatic patients (heartburn) or those with a history of restrictive surgery. In our series, only 4 out of 22 patients presented with preoperative symptoms of GERD, while the other 18 patients were completely asymptomatic. The diagnosis of HH was made preoperatively in only four patients; one presented with typical GERD, and three had a history of gastric banding. For these reasons, we think that it is important to standardize the OAGB with the Nissen-valve technique in order to give the surgeon an alternative in the event of intraoperative discovery of an HH, which happened in the vast majority of cases in this series despite a complete preoperative assessment.

It is impossible to know whether the patients in our series would have developed postoperative BR if we had not treated HH with the creation of the Nissen valve. Winstanley et al. recently reported a series of 63 patients who had standard OAGB in case of associated HH [18]. Although 10 of these patients had symptomatic preoperative GERD, none of them developed postoperative BR, with an average follow-up of 24 months. On the other hand, 20.6% of patients developed de novo GERD, requiring long-term PPI treatment. The type and size of HH were not specified in this study, and, above all, patients with a previous history of gastric banding were excluded from the study. This last point is important because the history of gastric banding has been identified by several studies as a risk factor for developing BR after OAGB. Poublon et al. compared the results of Roux-en-Y gastric bypass with those of OAGB in the event of prior failure of restrictive surgery [6]. They found a significantly higher rate of disabling reflux after OAGB (0.3% vs. 5.4%). Probably, the anatomical variations at the level of the cardia caused by the gastric banding could facilitate the onset of a BR, especially when an HH remains unrecognized during the realization of the OAGB. In an interesting study, Borbély et al. assessed 47 patients with symptomatic GERD after Roux-en-Y gastric bypass [19]. The presence of HH and in particular type II and III HH was frequently found, but other functional abnormalities were also reported, such as esophageal motility disorders, hypotonia, or transient relaxation episodes of the low esophageal sphincter. All these esophageal functional disorders are characteristic sequelae of gastric banding. In our series, three out of four patients with a previous history of gastric banding had typical GERD before the conversion. Two of them presented with persistent postoperative GERD, well controlled by long-term low-dose proton pump inhibitor intake, but none of these patients with a history of gastric banding developed postoperative BR. In our daily practice, we perform preoperative manometry and pH-metry in patients with previous gastric banding in case of evidence of esophageal dilation and dysmotility, detected during endoscopy or after UGI series. In these patients, if esophageal dysmotility is confirmed, we prefer to perform a low-pressure technique like laparoscopic Roux-en-Y gastric bypass (LRYGB).

The distinction between BR, acid reflux, and “mixed” reflux has not yet been extensively studied in patients with OAGB. Nehmeh et al., in a very recent study, reported a series of 43 patients who presented with symptomatic reflux after OAGB [20]. According to the results of the 24 h impedance pH monitoring, acid reflux was as frequent as BR and more frequent than mixed reflux (30.2% vs. 27.9% vs. 11.6%). The OAGB technique consists of making a long and narrow gastric pouch, and consequently the possibility of an important amount of acid production by the gastric mucosa is still present. For this reason, the association of OAGB and the presence of an untreated HH could favor the appearance of not only BR but also acid reflux or mixed reflux. In this sense, OAGB and LRYGB seem to have similar behavior, as shown by Madalosso et al. [21]. The authors reported in their series of patients operated on with a Roux-en-Y gastric bypass a 30% prevalence of HH, and its existence was a predictive factor for acid reflux after the operation. In the presence of a poorly continent cardia due to an HH, the purpose of the creation of a 360-degree wrap is to prevent the reflux of gastric liquid, independent of its quality and quantity.

In recent years, the creation of a Nissen fundoplication to prevent postoperative reflux after sleeve gastrectomy has been proposed by some surgical teams with good results [22]. Its systematic use could theoretically significantly reduce the incidence of de novo GERD after sleeve gastrectomy, which in some series reaches discouraging rates. In the present study, patients underwent this procedure to remedy a factor favoring reflux, such as HH. It was absolutely not our intention to consider OAGB with the Nissen valve as a technique to be adopted in a systematic way to prevent bile reflux after OAGB, which occurs in significantly less important percentages. Moreover, there is an important difference in the functional role of the valve between the two modified Nissen techniques. In the case of the Nissen sleeve, the antireflux valve is not excluded but is an integral part of the esophageal-gastric transit. On the other hand, in the OAGB, the 360-degree wrap works like a control mechanism, external to the pouch, that is supposed to increase the intragastric pressure in the area of the cardia, facilitating its continence and reducing the risk of HH recurrence. For this reason, a too loose antireflux valve would not systematically prevent bile from rising into the esophagus, and it could explain the appearance of postoperative BR even when the valve is well positioned and has not migrated into the mediastinum.

## 5. Conclusions

OAGB with a Nissen antireflux valve seems to be a safe and effective surgical technique in cases of patients with morbid obesity and HH. Although the operative time was longer compared to conventional OAGB, the rate of early postoperative complications was not affected. Moreover, the weight loss was not different from that expected after the realization of a standard OAGB. The functional results are encouraging even at the start of the experience, but the data presented are largely insufficient to establish its routine use to prevent BR after OAGB. On the other hand, it can be an extra arrow in the quiver of a surgeon who is faced with HH in a patient scheduled for OAGB.

## Figures and Tables

**Figure 1 jcm-11-06441-f001:**
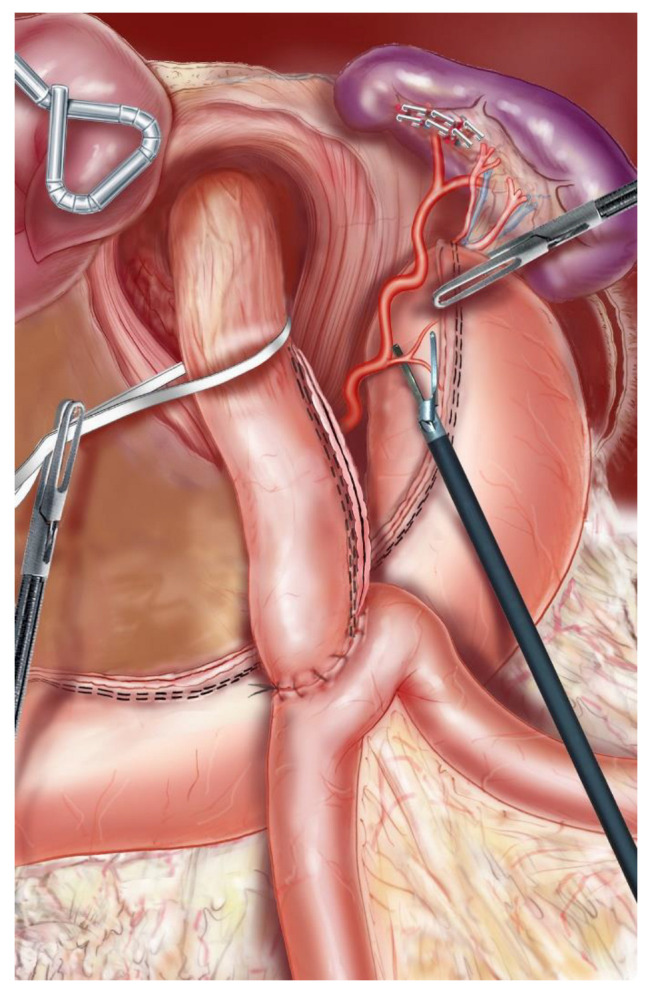
Dissection of the abdominal esophagus. In order to correctly mobilize the fundus, the posterior tuberosity artery and the last 2 short vessels should be systematically sectioned.

**Figure 2 jcm-11-06441-f002:**
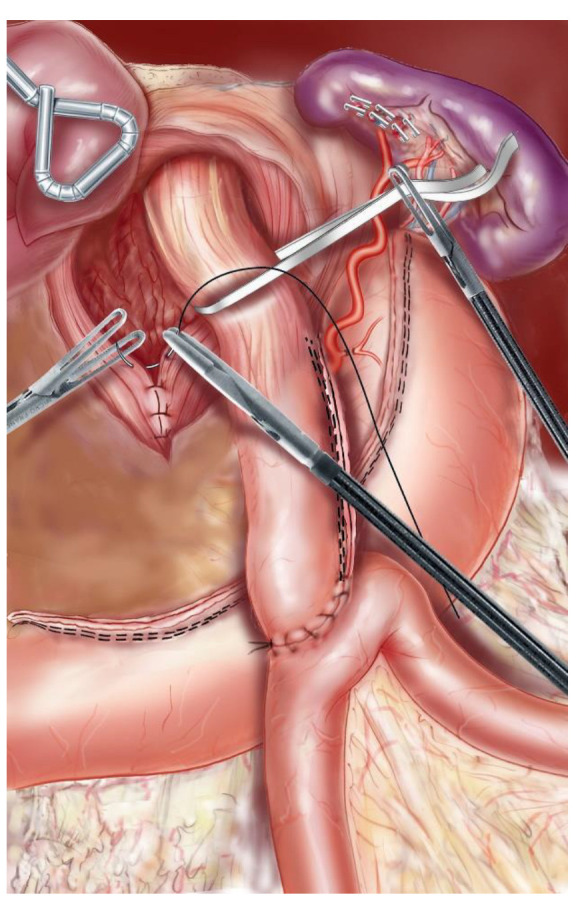
The hiatus was closed with 2–3 separate stitches of nonabsorbable sutures.

**Figure 3 jcm-11-06441-f003:**
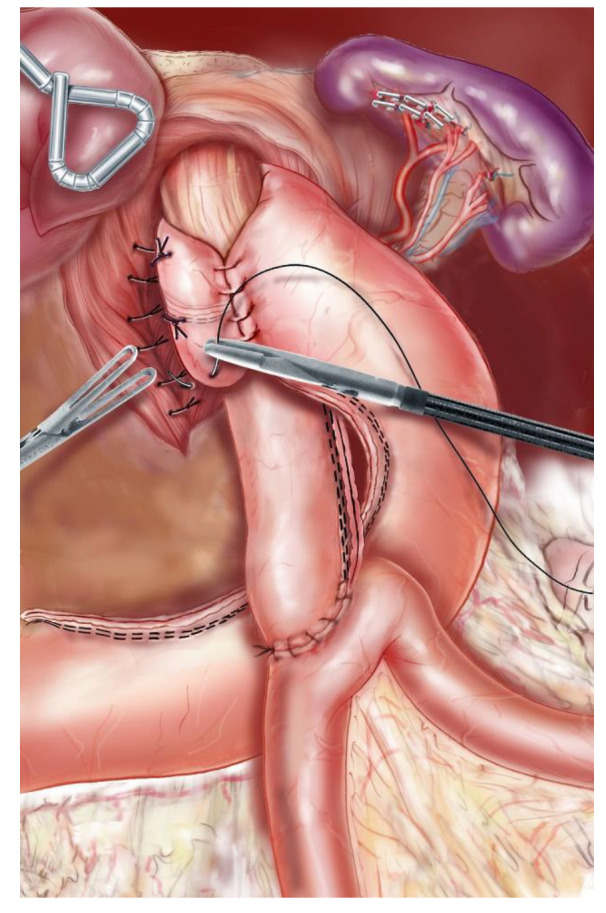
A 2.5–3 cm-long 360-degree wrap was created around the abdominal esophagus (Nissen technique).

**Table 1 jcm-11-06441-t001:** Clinical characteristics of the patients.

Characteristics	OAGB-Nissen (*n* = 22)
Age (years)	48 ± 10
Women/men	21/1
Preoperative weight (kg)	96 ± 9
Preoperative BMI (kg/m^2^)	40 ± 14
Type 2 diabetes	1 (4.5%)
Arterial hypertension	3 (13.6%)
Sleep apnea syndrome	5 (22.7%)
Symptomatic GERD	4 (18.2%)
Preoperative hiatal hernia	4 (18.2%)
Paraesophageal HH (type II)	2 (9.1%)
Paraesophageal and sliding HH (type III)	2 (9.1%)
History of bariatric surgery	5 (22.7%)
Gastric banding	4 (18.2%)
Intragastric balloon	1 (4.5%)

OAGB: one-anastomosis gastric bypass; BMI: body mass index; HH: hiatal hernia; GERD: gastroesophageal reflux disease.

**Table 2 jcm-11-06441-t002:** Intraoperative and postoperative data.

Characteristics	OAGB-Nissen (*n* = 22)
Operative time (minutes)	60 ± 20
Biliary loop length (mt)	
150 cm loop	11 (50%)
200 cm loop	11 (50%)
Associated cholecystectomy	1 (4.5%)
Hospital stay (days)	2 ± 1
Intraoperative bleeding	0
Early complications	0
Late complications	1 (4.5%)
Rehospitalization	0

## Data Availability

The data presented in this study are available upon request from the corresponding author (Sergio Carandina; sergio.carandina@gmail.com).

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
