# Peer review of "One-Anastomosis Gastric Bypass and Hiatal Hernia: Nissen Fundoplication with the Excluded Stomach to Decrease the Risk of Postoperative Gastroesophageal Reflux"

_jcm, 2022, doi:10.3390/jcm11216441_

Round 1
Reviewer 1 Report
it is a nice procedure and could be done, i have few comments please:
1. You talked about big HH type II or Type III, can you tell me please; why you could not detect it via Gastroscopy preoperatively?
2. the aim of doing fundoplication is to relief the GERD symptoms, but in your study, you have only 4 patients with GERD, so i don't understand why fundoplication and not Gastropexy. If you aim to reduce BR in these patients, i don't understand why you aren't doing gastric bypass?
3. The type of Follow up, how and when is not clear. how did you controlled your patients and when did you recommend the investigations and for who?
Author Response
Reviewer n.1
it is a nice procedure and could be done, i have few comments please:
- You talked about big HH type II or Type III, can you tell me please; why you could not detect it via Gastroscopy preoperatively?
Response: We thanks the Reviewer for its positive comments. In the present study, all the type II and III HH discovered both preoperatively and intraoperatively were hernias larger than 2 cm. Intraoperatively it’s difficult to estimate exactly the HH width but in all cases the pillar consistency and the content of HH sac necessitated a Nissen valve creation in order to decrease the risk of gastric pouch intrathoracic migration. Gastroscopy is essential in the preoperative preparation of the bariatric patients but its specificity and sensitivity for HH diagnosis is 66% and 72 % respectively (Hiatal hernia diagnosis prospectively assessed in obese patients before bariatric surgery: accuracy of high‐resolution manometry taking intraoperative diagnosis as reference standard. Santonicola et all., Surgical endoscopy 2020). Moreover, the preoperative workup and preparation of patients in our country can take a lot of time and endoscopy sometimes is performed several months before surgery. In this condition, a less 2 cm HH could also increase its dimensions with the risk to find intraoperatively a 3-4 cm HH.
- the aim of doing fundoplication is to relief the GERD symptoms, but in your study, you have only 4 patients with GERD, so i don't understand why fundoplication and not Gastropexy. If you aim to reduce BR in these patients, i don't understand why you aren't doing gastric bypass?
Response: Thanks to the Reviewer for this question. The aim of fundoplication is not only to relief GERD symptoms but also decrease significantly the risk of intrathoracic gastric migration and hernia recurrence. Gastropexy is a valid option but with a higher risk of hernia recurrence especially in case of large hiatal hernia. Moreover, the aim was not to reduce BR but to prevent the impairment of the gastroesophageal junction due to the presence of HH. The functional alteration of this region is described in several series as a factor that increase the risk of BR or acid reflux after bariatric surgery. We agree with the Reviewer that RYGB is the best option in case of severe preoperative GERD, but patients (3 patients) in this series who presented with these signs did not want RYGB procedure mainly because of the fear of postoperative bowel obstruction. For this reason, patients with preoperative GERD were also scheduled to be operated with this technique, while in the case of the intraoperative discovery of HH we preferred to guarantee the patients
The type of Follow up, how and when is not clear. how did you controlled your patients and when did you recommend the investigations and for who?
Response: we apologize to the Reviewer and we have added an explanation regarding the pattern of our follow-up in the MM section.

Reviewer 2 Report
One anastomosis GB and HH: Nissen fundoplication with the excluded stomach to decrease the risk of postop GERD
The current gold standard management of the obese patient with gerd (or at high risk for developing it) is a gastric bypass. Whether to perform a hiatal hernia repair in the asymptomatic patient in the setting of another weight reducing operation largely will depend on the size of the hernia itself and its consequent risk of sleeve or pouch herniation.
This study is limited by its retrospective nature and sample size. Additionally, I think the biggest drawback is that the authors performed an antireflux operation on all patients with a hiatal hernia, even those found incidentally in the operating room, whether reflux was present clinically or not. Thus, it is hard to determine the actual benefit of the anti reflux operation itself when patients did not have symptoms to begin with. Furthermore, only three patients out of 22 patients who received an antireflux operation had objective documentation of GERD-related changes. This is relevant because not all bariatric surgeons are equally aggressive when dealing with hiatal hernias, specially those incidentally discovered ones in an asymptomatic patient. Also, authors do not describe the hernia size they encountered in their patient population, or if there were other patients with hernias that were not operated on (and the reason for not operating on them).
The most commonly reported outcomes measure is EWL (excess weight loss, rather than total weight loss), not total weight loss %.
Authors claim no morbidity from surgery occurred, whether this was early or late. However, de novo GERD or BR should be accounted for as a morbidity given that the patient had no symptoms prior to surgery, yet received an anti-reflux operation, and now has reflux despite claiming to have lost over 100% of total body weight, which is major contributor for reflux in the obese population.
Authors should describe all acronyms once in the manuscript, either with a legend or in-text. \
As the authors recognize, not having a comparison group in which patients with similar characteristics would have developed BR given that all HH were treated, whether symptomatic or asymptomatic is another limitation of the study.
We typically obtain manometric studies after a gastric band, specially if longstanding, given the risk for esophageal motility, and if present or if associated with reflux, we tend to offer a gastric bypass rather than a sleeve, or in this case, a OAGB. Authors should consider describing their approach to revisional bariatric surgery after a failed band.
Author Response
One anastomosis GB and HH: Nissen fundoplication with the excluded stomach to decrease the risk of postop GERD
The current gold standard management of the obese patient with gerd (or at high risk for developing it) is a gastric bypass. Whether to perform a hiatal hernia repair in the asymptomatic patient in the setting of another weight reducing operation largely will depend on the size of the hernia itself and its consequent risk of sleeve or pouch herniation.
This study is limited by its retrospective nature and sample size. Additionally, I think the biggest drawback is that the authors performed an antireflux operation on all patients with a hiatal hernia, even those found incidentally in the operating room, whether reflux was present clinically or not. Thus, it is hard to determine the actual benefit of the anti reflux operation itself when patients did not have symptoms to begin with. Furthermore, only three patients out of 22 patients who received an antireflux operation had objective documentation of GERD-related changes. This is relevant because not all bariatric surgeons are equally aggressive when dealing with hiatal hernias, specially those incidentally discovered ones in an asymptomatic patient. Also, authors do not describe the hernia size they encountered in their patient population, or if there were other patients with hernias that were not operated on (and the reason for not operating on them).
Response: we thank the reviewer for its fair and very constructive feedback. Concerning the retrospective nature and the small sample size of patients we completely agree with the Reviewer that they are two important limitations of the present report. On the other hand, all the series published in literature regarding the use of a Nissen technique associated to bypass, RYGB or OAGB, are very small. For instance, Motola et al showed le results of a retrospective series of anti-reflux procedures after RYGB but only 4 patients had a Nissen procedures (Motola D, et al; Anti-reflux procedures after Roux-en-Y gastric bypass; Arq Bras Dig 2021). Werapitya et al. showed a series of 12 patients operated on Nissen procedure after OAGB with the aim of treating BR and avoiding conversion to RYGB (Werapitiya SB, Ruwanpura SP, Coulson TR. Laparoscopic fundoplication using the excluded stomach as a novel management option for refractory bile reflux following one anastomosis gastric bypass; Obes Surg. 2022). In our series we described the results concerning 22 patients with HH operated on concomitant OAGB and Nissen procedure. At our knowledge the present series is the larger series in literature.
The aim of fundoplication is not only to relief GERD symptoms but also decrease significantly the risk of intrathoracic gastric migration and hernia recurrence. Gastropexy is a valid option but with a higher risk of hernia recurrence especially in case of large hiatal hernia. Moreover, the aim of the present study was not to reduce BR but to prevent the impairment of the gastroesophageal junction due to the HH. The functional alteration of this region is described in several series as a factor that increase the risk of BR or acid reflux after bariatric surgery. Instead, at the UGI series performed at one (100% of patients) et three postoperative years (40% of patients) we didn’t find any sign of hiatal hernia recurrence (We added the following phrase in the results section: No signs of HH recurrence were evidenced in the UGI control series performed at 1 and 3 years after surgery).
Effectively, if we take in consideration postoperative results, only 3 out 22 patients showed GERD-related changes. On the other hand, of the 4 patients who presented with preoperative GERD, 50% resolved the symptoms and the other 50% improved it. Furthermore, of the 18 patients without preoperative symptoms, 17 had no postoperative GERD signs. In this sense, even if logically the data are not sufficient to firmly affirm it, the addition of fundoplication permitted to improve GERD and to decrease the postoperative risk of reflux linked to the presence of HH.
In the present study, all the type II and III HH detected both preoperatively and intraoperatively were hernias larger than 2 cm. Intraoperatively it’s difficult to estimate exactly the HH width but in all cases the pillar consistency and the content of HH sac necessitated a Nissen valve creation in order to decrease the risk of gastric pouch intrathoracic migration. Gastroscopy is essential in the preoperative preparation of the bariatric patients but its specificity and sensitivity for HH diagnosis is 66% and 72 % respectively (Hiatal hernia diagnosis prospectively assessed in obese patients before bariatric surgery: accuracy of high‐resolution manometry taking intraoperative diagnosis as reference standard. Santonicola et all., Surgical endoscopy 2020). Moreover, the preoperative workup and preparation of patients in our country can take a lot of time and endoscopy sometimes is performed several months before surgery. In this condition, a less 2 cm HH could also increase its dimensions with the risk to find intraoperatively a 3-4 cm HH. Patients with large HH, diagnosed preoperatively, especially if associated with the presence of a severe GERD, were referred for RYGB with gastropexy. In all patients in the present study there was an important discrepancy between preoperative gastroscopy and intraoperative finding. In fact, either the presence of HH was not detected, or it was underestimated. For these reasons we continued with our OAGB scheduled plan and we added Nissen fundoplication in order to decrease the future risk of HH recurrence.
Eventually, we agree with the Reviewer that RYGB is the best option in case of severe preoperative GERD, but patients (3 patients) in this series who presented with these signs did not want RYGB procedure mainly because of the fear of postoperative bowel obstruction. For this reason, patients with preoperative GERD were also scheduled to be operated with this technique, while in the case of the intraoperative discovery of HH we preferred to guarantee the patients the technique that it had been established preoperatively and for which they had signed informed consent.
The most commonly reported outcomes measure is EWL (excess weight loss, rather than total weight loss), not total weight loss %.
Response: we apologize to the Reviewer because we make a mistake in Results section. In the very last years the measure of %EWL has been progressively replaced by the “change in BMI” and “% total weight loss”. As matter of that, the instructions for authors of Obesity Surgery journal, the official journal of IFSO, state that is mandatory to express weight loss with these two types of measure. For these reasons, although we calculated both %TWL and %EWL, we used only %TWL, but instead to report %TWL we reported the value of %EWL. We fixed the error and we added both methods to calculate weight loss and both results in the manuscript.
Authors claim no morbidity from surgery occurred, whether this was early or late. However, de novo GERD or BR should be accounted for as a morbidity given that the patient had no symptoms prior to surgery, yet received an anti-reflux operation, and now has reflux despite claiming to have lost over 100% of total body weight, which is major contributor for reflux in the obese population.
Response: we apologize to the Reviewer for the mistake. We corrected the error in table 2 and in the Results section.
Authors should describe all acronyms once in the manuscript, either with a legend or in-text.
Response: we apologize to the Reviewer for the inaccuracy, the entire manuscript was revised and acronyms described.
As the authors recognize, not having a comparison group in which patients with similar characteristics would have developed BR given that all HH were treated, whether symptomatic or asymptomatic is another limitation of the study.
Response: We agree with the reviewer that the lack of a control group is another limitation of the study. On the other hand, it is difficult to find a control group in this situation. In fact, in our opinion, it would not be a correct procedure not to treat a type II and III hiatal hernia larger than 2 cm during a bariatric procedure. It would perhaps be more logical to compare the Nissen procedure to a simple gastropexy to see if a fundoplication is really necessary or if it is not enough to simply suture the diaphragmatic pillars. Currently, we are retrospectively collecting data on OAGB and RYGB operated patients undergoing concomitant gastropexy in order to evaluate long-term reflux outcomes.
We typically obtain manometric studies after a gastric band, specially if longstanding, given the risk for esophageal motility, and if present or if associated with reflux, we tend to offer a gastric bypass rather than a sleeve, or in this case, a OAGB. Authors should consider describing their approach to revisional bariatric surgery after a failed band.
Response: we agree with the Reviewer regarding the usefulness and importance of manometric studies after gastric banding. Unfortunately, this examination is not performed ordinary in our hospital and it is not easy to obtain quickly outside. Furthermore, the cost of the examination is completely in the charge of the patient as well as being a poorly tolerated procedure. For all these reasons, only in exceptional cases, endoscopy and UGI series has been integrated with a manometry. Our obesity center is located in a particular region of the country where bariatric surgeons performed a lot of banding procedures since late 1990s. For this reason, nowadays, we are faced to a very important number of patients with previous gastric banding who need a conversion to another technique. In our daily practice we perform manometry and Ph-metry in patients with previous gastric banding, in case of evidence of esophageal dilation and dysmotility, detected during endoscopy or after UGI series. In these patients, if esophageal dysmotility confirmed, we prefer to perform low pressure technique like LRYGB. As advised by the Reviewer we described our approach to revisional surgery after GB in Discussion section.

Reviewer 3 Report
This report shows the efficacy of creating a Nissen valve to treat refractory biliary reflux after OAGB. But, I have a few questions.
1: This study is retrospective and the number of cases included is too small.
2: Although the surgery is described, it should be shown in a schema for better understanding.
3: Comparison should be made with and without Nissen fundoplication, and it is difficult to prove the usefulness of Nissen fundoplication in a single arm.
4: There is no scientific data showing that Nissen fundoplication prevents bile reflux; 24-hour impedance and pH monitoring are needed.
Author Response
Reviewer n.3
This report shows the efficacy of creating a Nissen valve to treat refractory biliary reflux after OAGB. But, I have a few questions.
1: This study is retrospective and the number of cases included is too small.
Response: we apologize to the Reviewer because we were not able to clearly explain our point of view. In the present report, we didn’t want to treat refractory biliary reflux after OAGB but instead, to prevent the impairment of the gastroesophageal junction due to the presence of HH. The functional alteration of this region is described in several series as a factor that increase the risk of BR or acid reflux after bariatric surgery. For this reason, we proposed a feasible and reproducible technique for patients scheduled for OAGB that present a type II or III HH, in order to decrease the potential risk of BR due to the HH.
Concerning the retrospective nature and the small sample size of patients we completely agree with the Reviewer that they are two important limitations of the present report. On the other hand, all the series published in literature regarding the use of a Nissen technique associated to bypass, RYGB or OAGB, are very small. For instance, Motola et al showed le results of a retrospective series of anti-reflux procedures after RYGB but only 4 patients had a Nissen procedures (Motola D, et al; Anti-reflux procedures after Roux-en-Y gastric bypass; Arq Bras Dig 2021). Werapitya et al. showed a series of 12 patients operated on Nissen procedure after OAGB with the aim of treating BR and avoiding conversion to RYGB (Werapitiya SB, Ruwanpura SP, Coulson TR. Laparoscopic fundoplication using the excluded stomach as a novel management option for refractory bile reflux following one anastomosis gastric bypass; Obes Surg. 2022). In our series we described the results concerning 22 patients with HH operated on concomitant OAGB and Nissen procedure. At our knowledge the present series is the larger series in literature.
2: Although the surgery is described, it should be shown in a schema for better understanding.
Response: according to your advice, we added 3 figures for better explaining the surgical technique.
3: Comparison should be made with and without Nissen fundoplication, and it is difficult to prove the usefulness of Nissen fundoplication in a single arm.
Response: the purpose of the study was not changing the present OAGB technique. In this case, we would agree with the Reviewer that a double arm prospective randomized study should be necessary. But, GERD after OAGB is relatively rare and, in our opinion, this complication can be prevented by a better selection of patients. The purpose of the present study was only to show a possible solution in a particular situation like patients with type II and III HH scheduled for OAGB. Moreover, in the majority of our patients HH was identified intraoperatively, when good decisions have to take more quickly than in preoperative period.
On the other hand, the usefulness of Nissen procedure in GERD treatment and its prevention is well established by several studies. Furthermore, fundoplication techniques are an essential part of the HH surgical treatment. As matter of that, Nissen fundoplication has been also proposed with interesting results, in the prevention of GERD after sleeve gastrectomy (Aiolfi A, et al; Laparoscopic sleeve-fundoplication for morbidity obese patients with gastroesophageal reflux: systematic review and meta-analysis. Obes Surg 2021).
4: There is no scientific data showing that Nissen fundoplication prevents bile reflux; 24-hour impedance and pH monitoring are needed.
Response: again, we absolutely agree with the Reviewer. In the present study, the effectiveness of the technique was evaluated only clinically. Bile reflux when present after OAGB is extremely symptomatic, for this reason we have relied heavily on clinical signs. On the other hand, the only way to assess precisely the extent and quality of reflux remains the 24 H impedance and pH monitoring. Unfortunately, this examination is not performed ordinary in our hospital and it is not easy to obtain quickly outside. Furthermore, the cost of the examination is completely in the charge of the patient as well as being a poorly tolerated procedure. For all these reasons, only in exceptional cases, endoscopy has been integrated with a Ph-metry. However, the exact nature of the reflux can also be assessed on the basis of the response to treatment. Reflux that responds to PPIs is likely to have acidic origin, while patients who do not respond to such medications almost certainly suffer from pure bile reflux.
